# Effect of Rare Earth Elements (Y, La) on Microstructural Characterization and Corrosion Behavior of Ternary Mg-Y-La Alloys

**DOI:** 10.3390/ma16145141

**Published:** 2023-07-21

**Authors:** Mohamed Ali Ibrahim Alwakwak, Ismail Esen, Hayrettin Ahlatcı, Esma Keskin

**Affiliations:** 1Mechanical Engineering Department, Karabuk University, Karabuk 78050, Turkey; mohamedalwakwak1975@gmail.com; 2Metallurgical and Materials Engineering Department, Karabuk University, Karabuk 78050, Turkey; hahlatci@karabuk.edu.tr (H.A.); keskinesma5@gmail.com (E.K.)

**Keywords:** Mg, RE elements, corrosion behavior, microstructure

## Abstract

In this study, the microstructural properties and corrosion behavior of RE elements (Y, La) added to magnesium in varying minors after casting and homogenization heat treatment were investigated. Three-phase structures, such as α-Mg, lamellae-like phases, and network-shaped eutectic compounds, were seen in the microstructure results. The dendrite-like phases were evenly distributed from the eutectic compounds to the interior of the α-Mg grains, while the eutectic compounds (α-Mg + Mg) RE (La/Y)) were distributed at the grain boundaries. According to the corrosion results, the typical hydroxide formation for lanthanum content caused the formation of crater structures in the material, and with the increase in lanthanum content, these crater structures increased both in depth and in density. In addition, the corrosion products formed by Y_2_O_3_ and Y(OH)_3_ in the Mg-3.21Y-3.15 La alloy increased the thickness of the corrosion film and formed a barrier that protects the material against corrosion. The thinness of the protective barrier against corrosion in the Mg-4.71 Y-3.98 La alloy is due to the increased lanthanum and yttrium ratios. In addition, the corrosion resistance of both Mg-3.21Y-3.15 La and Mg-4.71 Y-3.98 La alloys decreases after homogenization. This negative effect on corrosion is due to the coaxial distribution of oxide/hydroxide layers formed by yttrium and lanthanum after homogenization.

## 1. Introduction

The opportunities that arise due to the day-to-day development of science and the increase in technology create an increase in the fields of application and therefore the evolution of the universe. A developing universe must be in a condition that is parallel to the needs of the rising population. For this reason, there is a need for more efficient, serial, and technological materials with equipment (optional) depending on the buyer’s request that can fulfil the requirements. While realizing all these, no ecological balance and permanent deformation disruption should occur. Materials made of light materials are impressive in appearance and have optimum success in the environment and conditions in which they are used by reducing energy consumption. In addition to fuel saving in ships, trains, airplanes, and automobiles, where efficiency is a priority and a weight of one thousand kilograms is obtained, light materials used in energy sources such as solar, hydrogen, and electricity reduce the emission of gases that may adversely affect the ecological environment and ecological balance, so more alternative lightweight materials have been used in enterprises [1,2,3]. 

Magnesium (Mg) stands out among metals due to its low density of 1.74 g/cm^3^ and its lower weight than steel and aluminum. Magnesium’s low density, excellent vibration resistance, strong damping, and high specific strength attributes have led to its widespread adoption across a variety of industries, including the military, medicine, electronics, and transportation [4,5,6,7]. However, because of their low-standard electrochemical potential (−2.37 V), Mg-based alloys usually have worse corrosion resistance than SHEs (standard hydrogen electrodes) [8] in the high-chloride environment of physiological systems. As a result, their usage is limited in a variety of applications [9,10,11]. Galvanic corrosion has been linked to poor corrosion resistance of Mg-based alloys, inhomogeneous distribution of second phases, or the presence of impurity elements, including Fe, Ni, Cu, and Co. This is brought on by a hydroxide coating (MgOH)_2_ that has developed on a Mg surface that is unstable [9,10]. Improvements to the microstructure [12,13,14], texture creation [15,16], and alloying [17,18,19] have all been employed to enhance magnesium’s low corrosive qualities [20,21,22]. There have been several thorough and in-depth basic analyses and research on the corrosion issue with magnesium alloys. People’s understanding of the corrosion behaviors and processes of magnesium alloys is growing deeper and more thorough. Research into magnesium and its alloys over the past decade has revealed that the addition of rare earth elements (RE) has remarkable effects on the structure and properties at room temperature and at elevated temperatures, sparking interest from the scientific and industrial communities [23,24,25]. Alloy purification with RE, grain refinement, structure-altering precipitate orientation, and the formation of novel intermetallics are responsible for these changes [26,27]. When RE elements are added to magnesium, a hard eutectic phase is created, which boosts the strength of the alloy and makes the metal more malleable [28,29,30,31]. Researchers are always looking for ways to improve the corrosion resistance of magnesium alloys by studying the effects of different alloying components on corrosion behavior [32,33]. The corrosion resistance of magnesium alloys can be greatly improved by using rare earth (RE) as an alloying element [34,35,36]. Nakatsugawa et al. [37], who studied the corrosion behavior of several magnesium alloys containing rare earth elements, discovered that the addition of rare earth elements improved the magnesium alloys’ resistance to corrosion. The better corrosion behavior of Mg alloys with RE elements may be due to the rare earth elements stabilizing the corrosion product, as suggested by Kiryuu et al. [38]. Using hydrogen evolution and electrochemical techniques, Liu et al. [39] assessed the impact of Ce and La addition to an AM60 magnesium alloy on corrosion behavior. By eliminating micro-galvanic couplings, they discovered that adding RE elements to an AM60 alloy increased its resistance to corrosion. After casting and rolling a ZM21 Magnesium (Mg) alloy, Goren et al. [40] looked into the microstructure, mechanical, and corrosion properties of La and Ca additions. The ZM21 + 0.5(La) Mg alloy has been shown to have superior corrosion resistance to the ZM21 Mg alloy during both casting and rolling. Corrosion behavior of Y addition to the AZ63 magnesium alloy in a 3.5% NaCl solution was studied by Li et al. [41]. They discovered that Mg-6%Al-3%Zn’s corrosion resistance was much enhanced by the addition of the yttrium alloy. Using a 3.5% NaCl solution, Manivannan et al. [42] studied the corrosion behavior of a Mg-6Al-1Zn + XYttrium (X = 0.5, 1.0, and 1.5 wt.% Y) magnesium alloy with varying yttrium additions. Grain refinement of the -phase from a continuous mesh morphology to a tiny morphology was shown to be responsible for Y’s beneficial effects on the Mg-6Al-1Zn magnesium alloy’s corrosion resistance. Wang et al. [43] studied the microstructure and corrosion behavior of TX31 alloys while methodically examining the impacts of minor RE elements (Y, La). The addition of Y and La to TX31 alloys improved corrosion film resistance and slowed corrosion rates.

Researchers have shown in most studies [44,45,46,47,48] that rare earth (RE) elements can improve corrosion resistance for Mg alloys. However, the effect of lanthanum and yttrium additions on the corrosion behavior of pure magnesium has not been investigated. In addition, the fact that it has not been investigated in basic studies, such as microstructure and alloy phases, has created a deficiency in the literature. The aim of this study is to examine how lanthanum and yttrium alloying elements added to magnesium in varying proportions change the microstructural and corrosive properties of the alloy. By using an immersion test and electrochemical measurements, this work examined the synthesis of magnesium-having RE elements (La and Y) and its corrosion resistance against a 3.5 weight percent NaCl solution. Low-angle X-ray diffraction (XRD), an energy-dispersive spectrometer (EDS), and scanning electron microscopy (SEM) were used to examine the impact of RE element addition on the microstructure and corrosion product layer.

### The Novelty of the Study

Although rare earth (RE) elements, such as yttrium and lanthanum, in the magnesium series have an important advantage in improving corrosion resistance, the effect of lanthanum and yttrium additions to pure magnesium on the corrosion behavior has not been investigated. In addition, the fact that it has not been investigated in basic studies, such as microstructure and alloy phases, has created a deficiency in the literature. In this study, how the lanthanum and yttrium alloying elements added to magnesium in varying pro-portions change the microstructural and corrosive properties of the alloy was examined in detail and the results were reported. Thus, this study aimed to show high performance in marine vehicles and aviation areas where corrosion resistance of these magnesium alloys is desired.

## 2. Materials and Methods

An atmosphere-controlled melt-based low-pressure casting system was used to melt and cast the alloys. Molten metal was poured into a cylindrical mold. In the first stage of the casting process, 99.9% pure magnesium ingots were thrown into the ladle. To measure the internal temperature of the melting furnace, a double thermal O-ring was placed on the top cover of the furnace to prevent leakage. At 780 °C crucible temperature, the pieces prepared from the main materials containing the alloys to be added in the determined proportions were added to the crucible and mixed for 15 min. The main master alloys containing rare earth elements were Mg-30La and Mg-30Y. Then, the master Mg-30Zr alloy, which was heated in the determined pro-portions at 200 °C, was added to the melt and mixed. CO_2_ + 0.8 SF_6_ shielding gas was supplied to the melting furnace to cut off the relation of the crucible with the atmosphere during melting. The liquid metal pipe, which was heated to 700 °C in a different furnace, was immersed in the melt. The mold, which was heated to 300 °C on the pipe, was filled at 2 bar pressure. The chemical compositions of the obtained alloys are given in Table 1. 

Table 1 shows the X-ray fluorescence (XRF Rigaku ZSX Primus II, Tokyo, Japan) % by weight chemical composition of the alloys produced. X-ray diffractometry (XRD Rigaku Ultima IV, Tokyo, Japan) was used at 10–90° and 3°/min to figure out the alloy’s phases after casting and corrosion.

In the Mg-Y [49] and Mg-La [50] binary phase diagrams, the Mg-Y alloy’s eutectic transformation temperature was 566 °C, close to the alloy systems’ phase-decomposition temperature, while the Mg-La alloy’s was 548 °C. To achieve the best homogenization effect and to prevent overheating, the alloy’s homogenization temperature was kept 5–10 °C below the solidus curves. Particularly, a higher homogenization temperature was preferred to reduce homogenization time, increase production efficiency, and provide excellent homogenization. Homogenization was set at 525 °C. Ingot castings were covered in aluminum foil and stored in SiO_2_ + graphite sand at 525 °C for 8 h. Homogenized alloys were taken from the furnace and cooled in water. Picral etching (6 g picric acid, 5 mL acetic acid, 10 mL distilled water, and 100 mL ethanol) was employed to characterize the alloys’ microstructure following casting and homogenization. The phase structure was investigated with an optical microscope. For a detailed analysis of phase morphologies, Carl Zeiss Ultra Plus Gemini SEM and EDS were utilized. Brinell hardness was tested using 2.5 mm steel balls at 187.5 N. 

Immersion corrosion testing used 3/4 cylindrical specimens with a 14 mm diameter and length surface. The surface area calculation of each sample was calculated one by one, and weight measurements were made with a Precisa brand balance with 0.1 mg precision. Immersion corrosion lasted 3–72 h in 3.5% NaCl. Every hour, chromic acid (180 g/L) and ethanol were used to clean the sample surface of corrosion residues. Chromic acid and ethanol obtained with 180 g CrO_3_ and 1 L distilled water were used to clean the corrosion residues formed on the sample surface at every hour interval. The corrosion samples were first kept in chromic acid for about 3 min and then pure water was used to clean the chromic acid in the sample. Corrosion samples cleaned with distilled water were kept in ethanol for about 2 min and dried. After the drying process, the individual weight measurements of the corrosion samples were made with precision scales and were left for immersion corrosion again. As a result of the immersion corrosion process, the average of the weight losses in grams measured at each hour interval was converted into milligrams. The surface area of the corrosion sample in mm^2^ was also converted to dm^2^. Thus, the milligram lost per square decimeter per hour was determined by dividing the average weight loss (mg) by the surface area (dm^2^). Weight losses were found by subtracting the pre-corrosion value from the post-corrosion value using these values. For example, the weight loss after 24 h was found by subtracting the weight/surface area value measured before corrosion from the weight/surface area value after 24 h (see Figure 7). The corrosion rate was calculated in mdd (milligrams per square decimeter per day (mg/dm^2^·day)). Corrosion rates per day were found by dividing weight losses by individual days. For example, since 72 h is 3 days, the weight loss after 72 h was divided by 3 to determine the corrosion rate corresponding to 3 days (see Figure 8). The computer-controlled DC105 Gamry model PC4/300 mA potentiostat/galvanostat device performed the potentiodynamic polarization test. Potentiodynamic polarization corrosion tested 3/4 cylindrical specimens wrapped in copper wire and cold-mounted. For the test, tapes with a surface area of 0.19 cm^2^ adhered to the surfaces of the samples, which were sanded using up to 2500 μm sandpaper, and the samples were left to the potentiodynamic polarization corrosion test. An electrolyte with a 3.5% NaCl concentration in distilled water was used as a corrosion medium. Electrochemical corrosion tests were performed in a standard three-electrode cell consisting of a graphite rod (CE) as a counter electrode, a saturated calomel electrode (SCE) as a reference electrode, and the sample as a working electrode with an open surface area of 1 mVs^−1^. After the electrochemical testing system was stable, scans were conducted at a rate of 1 mV/s^−1^ from 0.1 V versus open-circuit potential in a more noble direction up to 0.25 V versus the reference electrode. At least two samples were used to determine the corrosion properties in both immersion corrosion and potentiodynamic polarization corrosion tests.

## 3. Results and Discussion

### 3.1. XRD Patterns

XRD patterns of as-cast Mg-3.21Y-3.15 La and Mg-4.71 Y-3.98 La alloys are given in Figure 1a and Figure 1b, respectively. While the yttrium-rich Mg_24_Y_5_ phase is mostly seen in the Mg-3.21Y-3.15 La alloy in XRD standard cards, the MgY phase was observed in the Mg-4.71 Y-3.98 La alloy. In addition, LaMg_3_ phase peaks are frequently seen in both alloys in XRD standard cards. While the XRD peaks of the Mg-3.21Y-3.15 La alloy (Figure 1a) start with Mg_2_Y and La_2_Mg_17_ phases at 17° degrees, XRD peaks of the Mg-4.71 Y-3.98 La alloy (Figure 1b) started with α-Mg, Mg_12_La, and La_2_Mg_17_ at 14° degrees, and the peak of the Mg_12_La and Mg_2_Y phases occurred at 17° degrees. In the XRD analysis, the peak intensity was observed at 36.5° degrees, and this XRD peak showed α-Mg, Mg_2_Y, Mg_24_Y_5_, Mg_12_La, and La_2_Mg_17_ phases of the Mg-3.21Y-3.15 La alloy; meanwhile, α-Mg, Mg_12_La and La_2_Mg_17_ phases were observed in the Mg-4.71 Y-3.98 La alloy. While the XRD peaks of the Mg-3.21Y-3.15 La alloy end with the LaMg_3_ phase at 82° degrees (Figure 1a). XRD peaks of the Mg-4.71 Y-3.98 La alloy terminated with LaMg_3_ and La_2_Mg_17_ at 85° degrees (Figure 1b).

### 3.2. Microstructure

After the as-cast alloys, Figure 2a,b shows optical microscope images of the Mg-3.21Y-3.15 La and Mg-4.71Y-3.98 La alloys. Microstructures have three-phase structures. These include the eutectic compounds with a net structure and α-Mg phases that resemble lamellae. The eutectic compounds were mostly located near the grain boundaries of the α-Mg crystals, whereas the lamellae-like phases were uniformly dispersed throughout the crystals. The XRD (Figure 1a,b) peaks of these eutectic phases are spread to the grain boundaries, suggesting the presence of α-Mg + MgRE (La/Y) intermetallics. The eutectic phase (α-Mg + Mg_17_La_2_) exhibited a coarsening of its lamellae structure as the La concentration increased [51]. Dendrite-like phase structures and network structures at grain boundaries increased with Y content [52]. Figure 3a,b shows optical microscope pictures of Mg-3.21Y-3.15 La and Mg-4.71Y-3.98 La alloys following homogenization heat treatment. The homogenization procedure reduces the lamellae-like phases in the interior regions of the α-Mg grains, which helps distribute alloy components evenly in the alloy system. Zhang et al. [49] believe that the Mg_24_Y_5_ phase decreases at the grain borders after homogenization, while Guo et al. [53] believe that LaMg_12_ surrounds the α-Mg grains and separates them.

Figure 4a,b shows SEM micrographs of Mg-3.21Y-3.15 La and Mg-4.71Y-3.98 La alloys after casting. Table 2 shows the EDS analysis of the second phase with distinct morphologies labelled (A–F) in Figure 4a,b. There are Y and La phases dissolved in α-Mg at point A in the Mg-3.21Y-3.15 La alloy (Figure 4a), and it is obvious that point B is rich in Y. According to the XRD patterns (Figure 1a), the intermetallic, which appears as a tiny white hue in the grain and grain borders, is Mg_24_Y_5_, which has a high yttrium concentration. LaMg_3_ forms rod-shaped grain structures at position C in Figure 4a. In the Mg-4.71 Y-3.98 La alloy (Figure 4b), there are light white-contrast-colored cubic and rod-like intermetallics in the grain structure at point D. At this point, considering the % La content, it is thought that it may be the La_2_Mg_17_ phase, which is also present in the XRD patterns (Figure 1b). It may belong to Mg_24_Y_5_ and LaMg_3_ intermetallics of small white-colored structures in rod form located in the grain and grain boundaries at point E. At the F point, according to the ratios of La and Y, which are close to each other in percent, wherein the intermetallics may be Mg_12_La and Mg_2_Y, which are rich in magnesium. Figure 5a,b shows homogenized SEM micrographs of Mg-3.21Y-3.15 La and Mg-4.71Y-3.98 La alloys. Table 3 shows the EDS results of the second phases with varied morphologies (A–F) in Figure 5a,b. In the Mg-3.21Y-3.15 La alloy (Figure 5a), there are intermetallics with a small size and cubic appearance, which are heavily deposited at the grain boundaries at the A point. In addition to the structures at these grain boundaries, rectangular-shaped structures are also seen within the grain. For the intermetallics found here, it is thought that the intermetallics found in the XRD patterns (Figure 1a) may be Mg_24_Y_5_ and Mg_12_La. In addition to the rod-like structures in the grain at point B, small-sized white structures in the grain boundary lines attract attention. The structures located here are thought to be Mg_2_Y and LaMg_3_. At the C point, the elliptical structure with white contrast in the grain may belong to the Mg_2_Y intermetallic. The Mg-4.71 Y-3.98 La alloy (Figure 5b) has dendritic rod-like features at the D point. This structure extends from the grain to its edges. In addition, small and round-shaped structures are observed within the grain towards the grain boundaries. It is thought that the structure in dendritic form is LaMg_12_ and the round-shaped structure may be MgY. At point E, there are rod-like structures that extend from the grain boundary into the grain, and some of them are inside the grain. It can be assumed that these structures may be La_2_Mg_17_. At point F, the structure reminiscent of a triangle and the small cubic-shaped structure located at the corner of this structure draws attention. The intermetallics constituting this structure are thought to be Mg_12_La and Mg_2_Y.

### 3.3. Hardness Test Results

A comparison of the hardness results of the Mg-3.21Y-3.15 La and Mg-4.71 Y-3.98 La alloys after casting and homogenization is given in Figure 6. When the hardness results are examined in general, a slight increase in hardness is observed after the homogenization heat treatment. The reason for this is thought to be the homogeneous dispersion of the secondary phases. As can be seen in Figure 6, homogenized alloy Mg-3.21Y-3.15 La has the highest hardness, reaching 93.70 ± 1.75 HB, while homogenized alloy Mg-4.71 Y-3.98 La has the lowest hardness, reaching 87.37 ± 0.2 HB. In addition, the hardness measurements after casting are also parallel to those after homogenization.

### 3.4. Corrosion Behavior

#### 3.4.1. Immersion Tests

The change in weight loss of Mg-3.21Y-3.15 La and Mg-4.71 Y-3.98 La alloys after 24 h is shown in Figure 7; corrosion rates after 72 h are given in Figure 8 comparatively. While the alloys corroded after homogenization show more weight loss, alloys corroded after casting exhibited better corrosion behavior. The Mg-4.71 Y-3.98 La alloy showed negative behavior against corrosion both after homogenization and after casting. The Mg-3.21Y-3.15 La alloy showed better corrosion behavior compared to the Mg-4.71 Y-3.98 La alloy. In addition, the Mg-3.21Y-3.15 La and Mg-4.71 Y-3.98 La alloys showed more stable corrosion behavior after 12 h against the corrosion they were exposed to after casting and homogenization. The least weight loss between the Mg-3.21Y-3.15 La and Mg-4.71 Y-3.98 La alloys after 24 h was observed in the Mg-3.21Y-3.15 La alloy, which corroded after casting, and this value was 0.275971 ± 0.02 mg/dm^2^. The highest weight loss was observed in the homogenized Mg-4.71 Y-3.98 La alloy with a value of 0.583942 ± 0.01 mg/dm^2^. The Mg-3.21Y-3.15 La alloy, which corroded after casting, showed the lowest corrosion rate with a value of 0.307679 mg/(dm^2^·day) after 72 h.

The post-corrosion XRD patterns of the Mg-3.21Y-3.15 La and Mg-4.71 Y-3.98 La alloys corroded after casting are given in Figure 9a and Figure 9b, respectively. In XRD standard cards, mostly Y(OH)_3_ formed by yttrium and La (OH)_3_ formed by lanthanum oxide films were encountered. Considering the intensity/counts on XRD standard cards for both alloys, it is seen that the peaks are more intense in the Mg-4.71 Y-3.98 La alloy. While the XRD peaks of the Mg-3.21Y-3.15 La alloy (Figure 9a) started at 16° after corrosion, after corrosion of the Mg-4.71 Y-3.98 La alloy (Figure 9b), XRD peaks started at 17°. In these two alloys, the post-corrosion XRD peaks were La_2_O_3_, La (OH)_3_, La_2_MgO_x_, Mg (OH)Cl, and Y(OH)_3_. In XRD analysis, the peak intensity was observed at 36°, and MgO, La_2_MgO_x_, and La (OH)_3_ peaks were observed in these two alloys. In addition, Y_2_O_3_ peaks were observed in the Mg-3.21Y-3.15 La alloy, and La_2_O_3_ peaks were observed in the Mg-4.71 Y-3.98 La alloy. While the XRD peaks of the Mg-3.21Y-3.15 La alloy ended with a Y(OH)_3_ oxide film at 85° (Figure 9a), XRD peaks of the Mg-4.71 Y-3.98 La alloy terminated with Y_2_O_3_, Y(OH)_3_, and La (OH)_3_ at 87° (Figure 9b). The post-corrosion XRD patterns of the Mg-3.21Y-3.15 La and Mg-4.71 Y-3.98 La alloys corroded after homogenization are given in Figure 10a and Figure 10b, respectively. While La_2_MgO_x_ oxide film was seen in the Mg-3.21Y-3.15 La alloy on XRD standard cards, the La (OH)_3_ peak was observed in the Mg-4.71 Y-3.98 La alloy. In addition, Y(OH)_3_ oxide films formed with yttrium were found in both alloys in XRD standard cards. Considering the density/counts on XRD standard cards for both alloys, there was a similar situation to that in post-casting corrosion (Figure 9). While the post-corrosion XRD peaks of the Mg-3.21Y-3.15 La alloy (Figure 10a) started at 16°, after corrosion of the Mg-4.71 Y-3.98 La alloy, XRD peaks (Figure 10b) started at 15°. In these two alloys, the post-corrosion XRD peaks were La (OH)_3_, La_2_MgO_x_, Mg (OH)Cl, and Y(OH)_3_. In addition, the La_2_O_3_ peak was also seen in the Mg-3.21Y-3.15 La alloy. In XRD analysis, peak intensity was observed at 36°, and MgO and La (OH)_3_ peaks were observed in these two alloys. In addition, La_2_O_3_ peaks were observed in the Mg-3.21Y-3.15 La alloy, and Y_2_O_3_ and La_2_MgO_x_ peaks were observed in the Mg-4.71 Y-3.98 La alloy. While the XRD peaks of the Mg-3.21Y-3.15 La alloy ended with Y(OH)_3_, Y_2_O_3_, MgO, La_2_O_3_, and La_2_MgO_x_ oxide films at 82° (Figure 10a), XRD peaks of the Mg-4.71 Y-3.98 La alloy terminated with Y_2_O_3_ at 87° (Figure 10b).

Overall, carbon contamination is very similar for both casting and post-homogenization of Mg-3.21Y-3.15 La and Mg-4.71 Y-3.98 La alloys. The lanthanum content is minimal near the surface, resulting in a near-surface magnesium-dominated treatment. The oxygen distribution allows for a rough calculation of the corrosion layer thickness. The layers are much thinner with greater lanthanum concentrations. In addition, the increase in both solid dissolved yttrium and rich yttrium regions caused thinning of the corrosion layer. Many phase peaks appear during the corrosion process, complicating the assessment. One of them is lanthanum-weighted peaks. The α-Mg and a-Mg + La_2_Mg_17_ eutectic phases, which are found in the pre-corrosion XRD patents (Figure 1) and especially in the SEM images (Figure 4 and Figure 5), cause galvanic corrosion, while the Mg_12_La phase is thought to cause micro-galvanic corrosion [51,54]. In addition, the La_2_Mg_17_ phase is thought to form the La_2_MgO_x_ phase after corrosion [55]. Different oxidation states of magnesium emerged for varying lanthanum and yttrium weight contents, which is also seen in post-corrosion XRD patents (Figure 9 and Figure 10). Typical hydroxide formation [56] for low-lanthanum content in Mg-3.21Y-3.15 La alloy caused the formation of crater structures in the material, and with the increase in lanthanum content, these crater structures increased both in their depth and in their densities. The presence of trace quantities of La_2_O_3_ and the subsequent production of a phase known as La_2_MgO_x_ prevented the synthesis of La(OH)_3_ in the Mg-3.21Y-3.15 La alloy [55,57]. During the corrosion process, another peak that complicated the evaluation is the phase peaks containing yttrium. When the addition of yttrium exceeded 2.5%, the secondary phase, Mg_24_Y_5_, was formed at the grain boundary (Figure 1), and a continuous barrier was formed to prevent corrosion that may occur in the region thanks to this phase [58]. In SEM micrographs (see SEM micrographs (Figure 4 and Figure 5)), the Mg_24_Y_5_ phase, which has the appearance of shiny pearls trapped in the grain boundaries, became more concentrated as the amount of yttrium increased. However, the intensity of this phase negatively affected the barrier against corrosion and increased the corrosion rate. Corrosion diffusion started with crater formation adjacent to the Mg_24_Y_5_ phase, followed by a concentrated corrosion attack in the yttrium-rich regions, and finally pitting occurred with the affinity of the α-Mg phase with oxygen [59]. The α-Mg phase of the samples submerged in the 3.5% NaCl solution was the initial site of corrosion. Furthermore, chlorine has a significant impact on corrosion’s development. The films are permeable to chlorine oxide and hydroxide; therefore, the corrosion interface is reached [60,61]. Therefore, it is thought that the Mg(OH)Cl peak seen in the XRD patents (Figure 9 and Figure 10) after corrosion affected corrosion negatively. In addition, corrosion products formed by yttrium with oxygen (Y_2_O_3_) and hydroxide (Y(OH)_3_) in the Mg-3.21Y-3.15 La alloy increased the density of the corrosion film and formed a barrier that protected the material against corrosion. Therefore, Y_2_O_3_ and Y(OH)_3_ phase formations slowed down the corrosion rate [43]. This corrosion behavior is similar to that of the Mg-4.71 Y-3.98 La alloy, but the protective layer thickness was in a thin state. The thinness of the protective barrier against corrosion in the Mg-4.71 Y-3.98 La alloy was attributed to increased lanthanum and yttrium ratios. In addition, there was a decrease in corrosion resistance in both the Mg-3.21Y-3.15 La and Mg-4.71 Y-3.98 La alloys after homogenization. It is thought that this negative effect on corrosion is due to the coaxial distribution of the oxide/hydroxide layers formed by yttrium and lanthanum after homogenization.

Figure 11a,b shows SEM micrographs of the Mg-3.21Y-3.15 La and Mg-4.71Y-3.98 La alloys immersed in corrosion after casting. Table 4 shows the EDS analysis of the second phases with distinct morphologies labelled (A-D) in Figure 11a,b. When we look at the SEM micrographs (Figure 11) corroded after casting, it can be seen that the presence of pit-shaped corrosion surfaces and porosity structures is increased in the Mg-4.71 Y-3.98 La alloy compared to the Mg-3.21Y-3.15 La alloy. While hydroxide formations are generally found in Mg-4.71 Y-3.98 La alloys, it is thought that oxide formations are more common in Mg-3.21Y-3.15 La alloys. While the rough areas seen at point A in the Mg-3.21Y-3.15 La alloy (Figure 11a) are thought to be MgO and Y_2_O_3_ phases, crater formations at point B are thought to be the La_2_O_3_ corrosion phase. In the Mg-4.71 Y-3.98 La alloy (Figure 11b), it is assumed that MgO, La (OH)_3_, and La_2_O_3_ intermetallics are formed in the pit-shaped corrosion zones at point C. At point D, Cl and H are completely dissolved in the structure. It is possible that the Mg (OH)Cl corrosion peak, which is also found in the XRD patents (Figure 9), may occur here. Besides, it is thought that there are oxide phases formed by lanthanum and yttrium at this point. Figure 12a,b shows SEM micrographs of the Mg-3.21Y-3.15 La and Mg-4.71Y-3.98 La alloys following homogenization and dip corrosion. Table 5 shows the EDS findings of the second phases with distinct morphologies labelled (A-D) in Figure 12a,b. Looking at the SEM micrographs (Figure 12) corroded after homogenization, it can be seen that the Mg-4.71 Y-3.98 La alloy has intensely indented structures, and the Mg-3.21Y-3.15 La alloy has small pits between the porous structures. The Mg-3.21Y-3.15 La and Mg-4.71 Y-3.98 La alloys have formations similar to the SEM micrographs (Figure 12) with corrosion after casting. While it is assumed that the Mg-3.21Y-3.15 La alloy (Figure 12a) has MgO, La_2_O_3_, and Y_2_O_3_ phases found in XRD patents (Figure 10) at point A, it is assumed that at point B there are MgO and mostly La_2_O_3_ phases. Due to the complete dissolution of hydrogen at the C point in the Mg-4.71 Y-3.98 La alloy (Figure 12b), it is thought that LaMgO_x_ and La_2_O_3_ phases are formed, as well as the La (OH)_3_ corrosion peak. At the D point, it is assumed that there are La_2_O_3_ and La (OH)_3_ peaks.

#### 3.4.2. Potentiodynamic Polarization (PD) Tests

Anodic polarization curves demonstrate magnesium dissolution and cathodic hydrogen evolution via water reduction [62]. Figure 13 shows current–voltage curves of the Mg-3.21Y-3.15 La and Mg-4.71Y-3.98 La alloys, while Table 6 lists corrosion characteristics. As can be seen in Figure 13 and Table 6, there is a strong decrease in corrosion current densities (I_corr_) of the Mg-3.21Y-3.15 La alloy compared to the Mg-4.71 Y-3.98 La alloy both after casting and after homogenization. Accordingly, as a result of the test, the Mg-3.21Y-3.15 La alloy shows the best corrosion behavior by creating -1.17 V corrosion potential at 220.00 μA/cm^2^ current density after casting; the Mg-4.71 Y-3.98 La alloy, on the other hand, showed the worst corrosion behavior by creating -1.30 V corrosion potential at 371.00 μA/cm^2^ current density after homogenization. Microstructure changes, which were more evident in XRD patents (Figure 1) and SEM micrographs (Figure 3 and Figure 5), which changed due to the addition of yttrium and lanthanum, created differences in corrosion resistance. The preceding section details the genesis of the corrosive phases and the potentiodynamic polarization curves favoring immersion corrosion.

Figure 14a,b show SEM micrographs of Mg-3.21Y-3.15 La and Mg-4.71Y-3.98 La alloys subjected to electro-chemical corrosion (potentiodynamic polarization) after casting. Table 7 shows the EDS results of the second phases with different morphologies (A-D) in Figure 14a,b. Looking at the SEM micrographs (Figure 14) corroded after casting, the Mg-4.71 Y-3.98 La alloy has a porosity surface that is more exposed to corrosion compared to the Mg-3.21Y-3.15 La alloy. There are hydroxide and oxide formations similar to the SEM micrographs (Figure 11) given after immersion corrosion in Figure 14. It has been observed that the Mg-3.21Y-3.15 La alloy (Figure 14a) has yttrium and lanthanum content in almost similar proportions at point A. At this point, in addition to MgO, oxide and hydroxide corrosion peaks formed by lanthanum and yttrium are likely to occur. At point B, there is a situation similar to point A. At the C point in the Mg-4.71 Y-3.98 La alloy (Figure 14b), since the hydrogen is completely dissolved in the structure, it is thought that La (OH)_3_ and Y(OH)_3_ phases may be formed. In addition, MgO, La_2_O_3_, and Y_2_O_3_ phases are likely to form at this point. At the D point, similar to point C, oxide phases formed by magnesium, yttrium, and lanthanum may be present. Figure 15a,b shows SEM micrographs of Mg-3.21Y-3.15 La and Mg-4.71Y-3.98 La alloys subjected to electro-chemical corrosion (potentiodynamic polarization) following homogenization. Table 8 shows the EDS analysis of the second phases with distinct morphologies labelled (A-D) in Figure 15a,b. Looking at the SEM micrographs (Figure 15) corroded after homogenization, it can be seen that the Mg-3.21Y-3.15 La alloy has a more porous surface than after casting (Figure 14a). In the Mg-4.71 Y-3.98 La alloy, the area exposed to corrosion has a rougher and denser porosity surface. In the Mg-3.21Y-3.15 La alloy (Figure 15a), it is assumed that the LaMgOx phase formed by lanthanum, magnesium, and oxygen forms at point A. Mg0, La_2_O_3_, and Y_2_O_3_ corrosion phases are likely to occur at point B. In the Mg-4.71 Y-3.98 La alloy (Figure 15b), there is a situation similar to the corrosion peaks at point B at points C and D.

## 4. Conclusions

The following results are given regarding the microstructural properties and corrosion behavior of RE elements (Y, La) added to magnesium in varying minor proportions after casting and homogenization heat treatment:
Three-phase structures, such as α-Mg, lamellae-like phases and network-shaped eutectic compounds, were seen in the microstructure results. The dendrite-like phases were evenly distributed from the eutectic compounds to the interior of the α-Mg grains, while the eutectic compounds (α-Mg + MgRE (La/Y)) were mostly distributed at the grain boundaries.The homogenized alloy Mg-3.21Y-3.15 La had the highest hardness with 93.70 ± 1.75 HB, while the homogenized alloy Mg-4.71 Y-3.98 La had the lowest hardness with 87.37 ± 0.2 HB.According to the immersion corrosion, after 24 h, there was a minimum weight loss of 0.275971 ± 0.02 mg/dm^2^ in the Mg-3.21Y-3.15 La alloy after casting; the highest weight loss was found in the homogenized Mg-4.71 Y-3.98 La alloy with a value of 0.583942 ± 0.01 mg/dm^2^. The Mg-3.21Y-3.15 La alloy, which corroded after casting, showed the lowest corrosion rate with a value of 0.307679 mg/(dm^2^·day) after 72 h.The formation of La (OH)_3_ caused the formation of crater structures in the material, and with the increase in lanthanum content, these crater structures increased both in their depth and in their densities. In the Mg-3.21Y-3.15 La alloy, a barrier was formed with Y_2_O_3_ and Y(OH)_3_ that protected the material against corrosion. The thinness of the protective barrier against corrosion in the Mg-4.71 Y-3.98 La alloy was attributed to the increased lanthanum and yttrium ratios. This negative effect on corrosion was due to the coaxial distribution of oxide/hydroxide layers formed by yttrium and lanthanum after homogenization.While it showed the best corrosion behavior by creating a corrosion potential of -1.17 ± 0.04 V at a current density of 220 ± 1.5 μA/cm^2^ after casting at Mg-3.21Y-3.15 La, Mg-4.71 Y-3.98 La, after homogenization, showed the worst corrosion behavior by creating a corrosion potential of -1.30 ± 0.02 V at a current density of 371 ± 1.2 μA/cm^2^.

## Figures and Tables

**Figure 1 materials-16-05141-f001:**
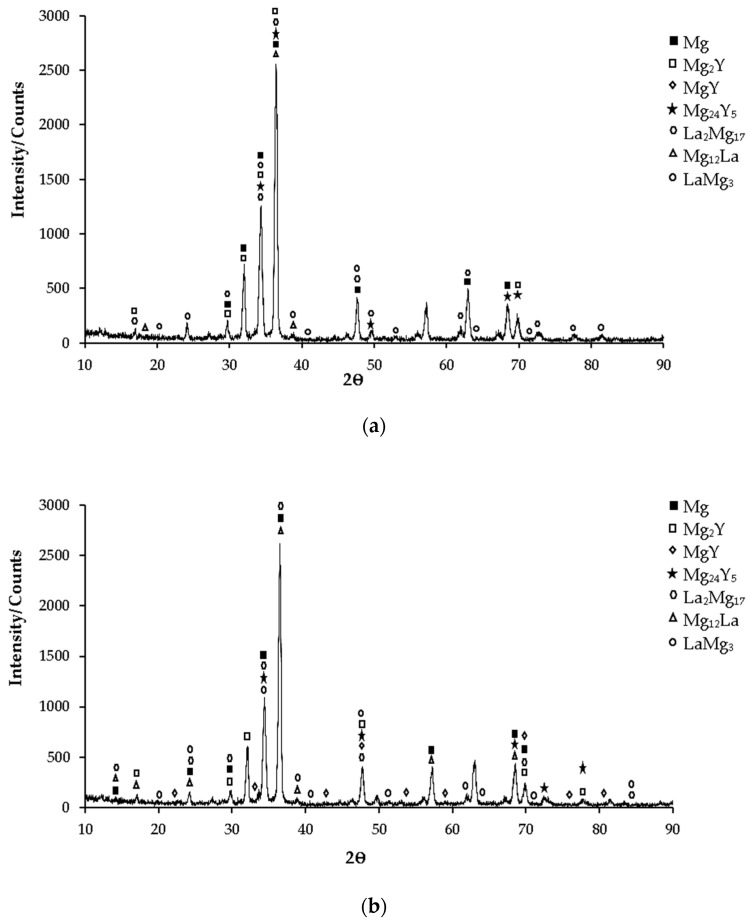
After-cast alloy XRD patterns: (**a**) Mg-3.21Y-3.15 La and (**b**) Mg-4.71 Y-3.98 La.

**Figure 2 materials-16-05141-f002:**
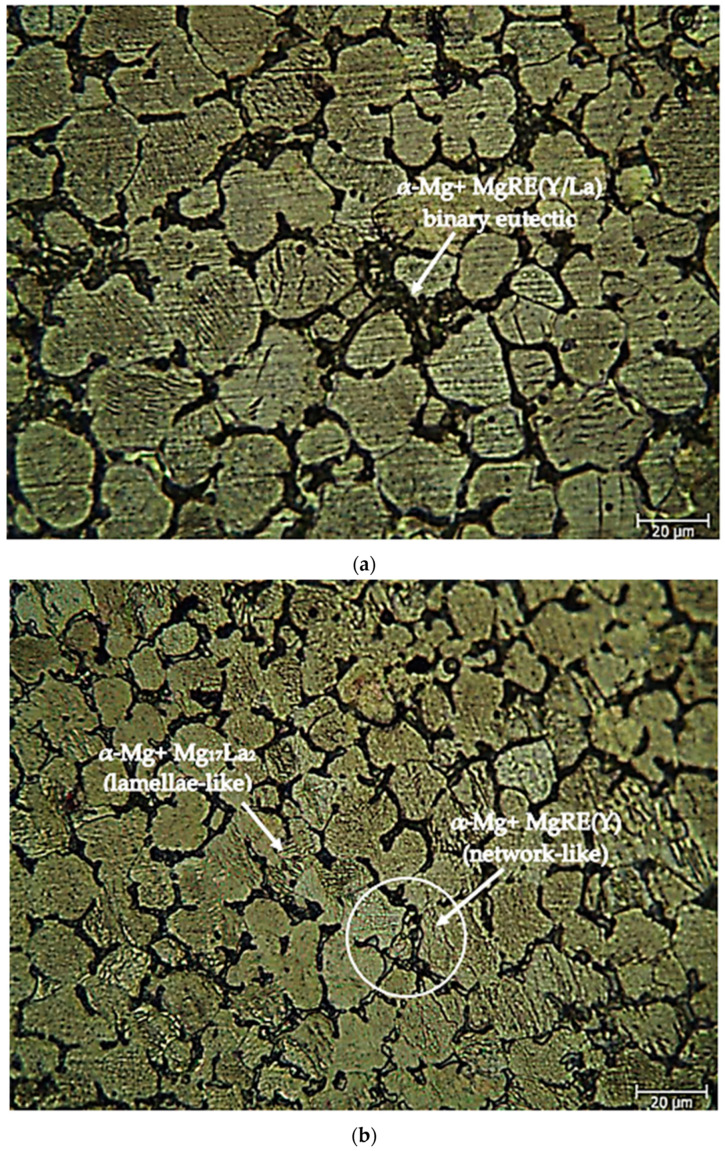
After-cast alloy optical microscope image at 50X: (**a**) Mg-3.21Y-3.15 La and (**b**) Mg-4.71 Y-3.98 La.

**Figure 3 materials-16-05141-f003:**
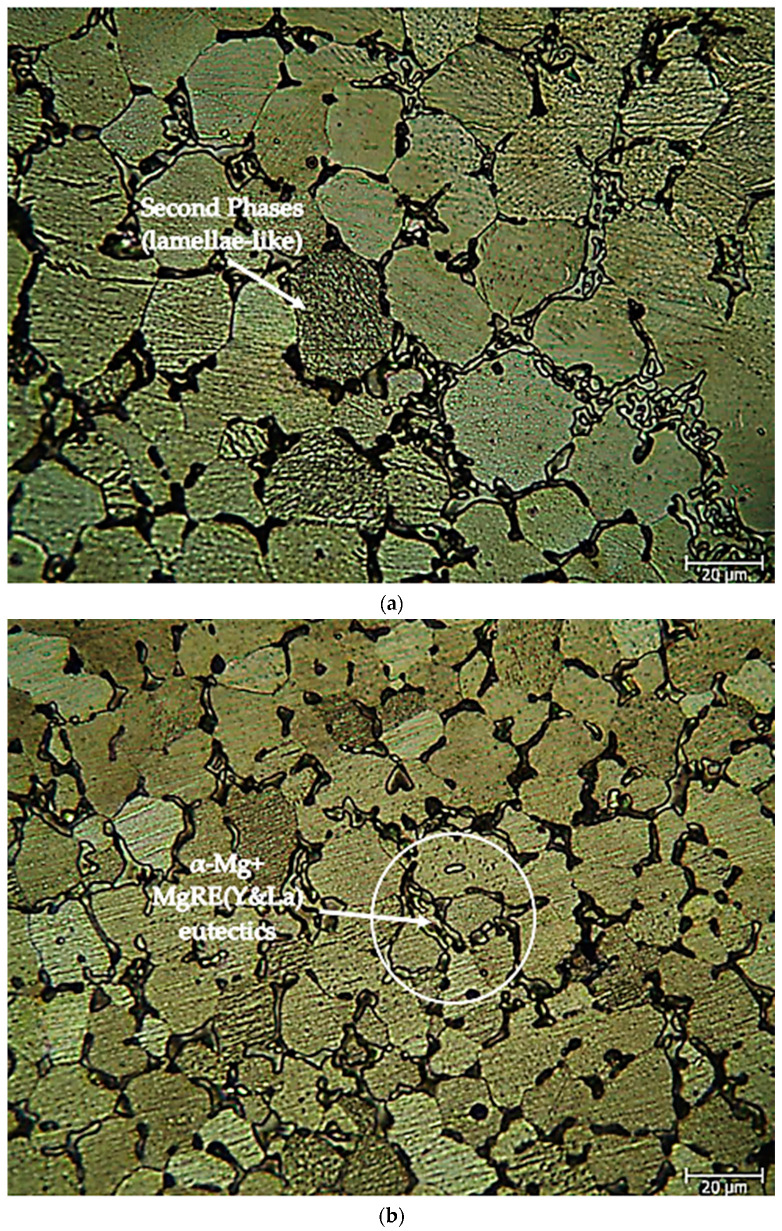
Homogenized alloy optical microscope images at 50X: (**a**) Mg-3.21Y-3.15 La and (**b**) Mg-4.71 Y-3.98 La.

**Figure 4 materials-16-05141-f004:**
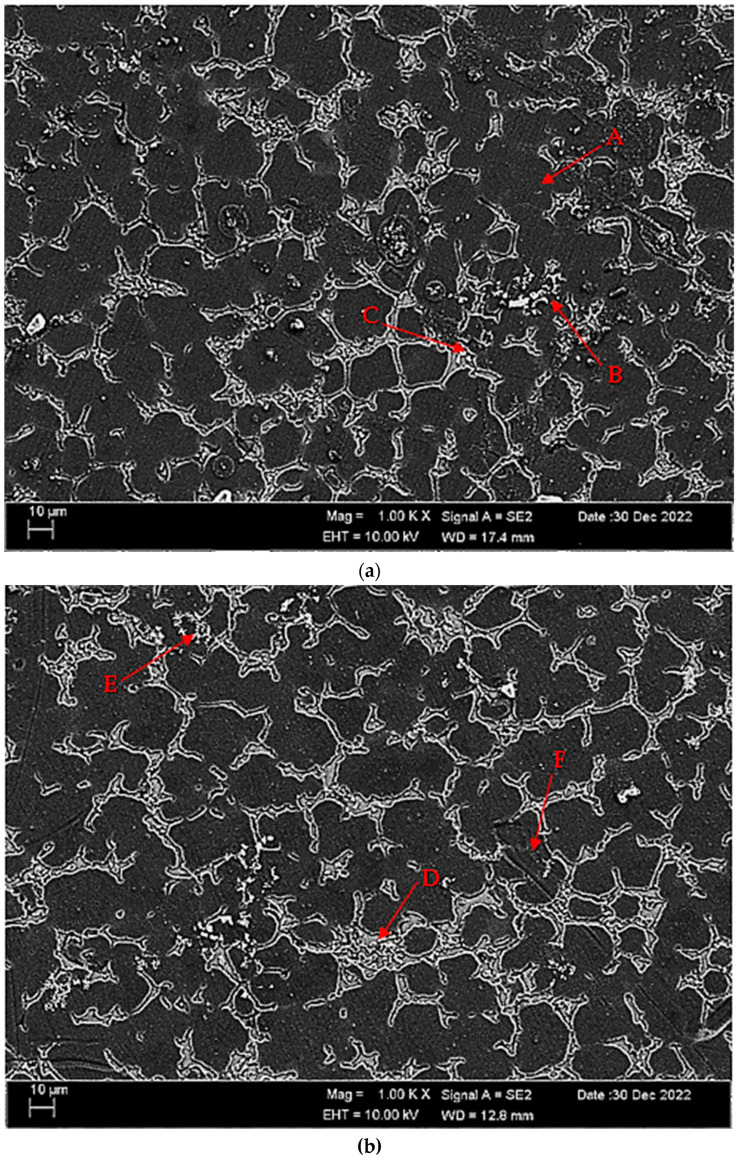
After-cast alloy SEM micrographs at 1kX: (**a**) Mg-3.21Y-3.15 La and (**b**) Mg-4.71 Y-3.98 La.

**Figure 5 materials-16-05141-f005:**
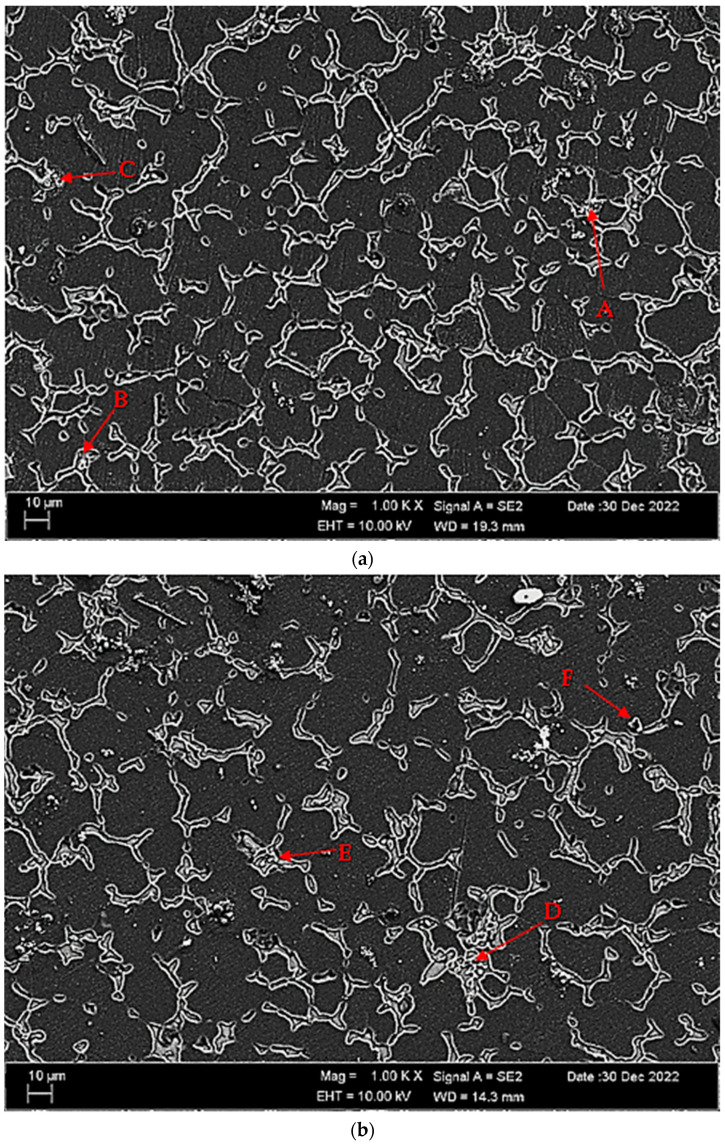
As-homogenized alloy’s SEM micrographs: (**a**) Mg-3.21Y-3.15 La and (**b**) Mg-4.71 Y-3.98 La.

**Figure 6 materials-16-05141-f006:**
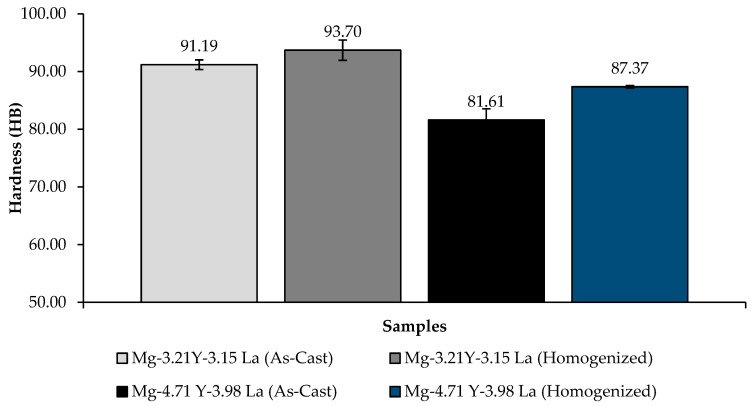
Hardness results of Mg-3.21Y-3.15 La and Mg-4.71 Y-3.98 La alloys after casting and homogenization.

**Figure 7 materials-16-05141-f007:**
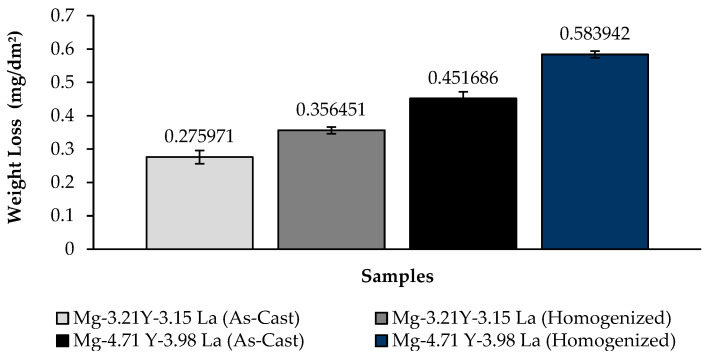
Results of immersion corrosion weight loss of Mg-3.21Y-3.15 La and Mg-4.71 Y-3.98 La alloys after casting and homogenization.

**Figure 8 materials-16-05141-f008:**
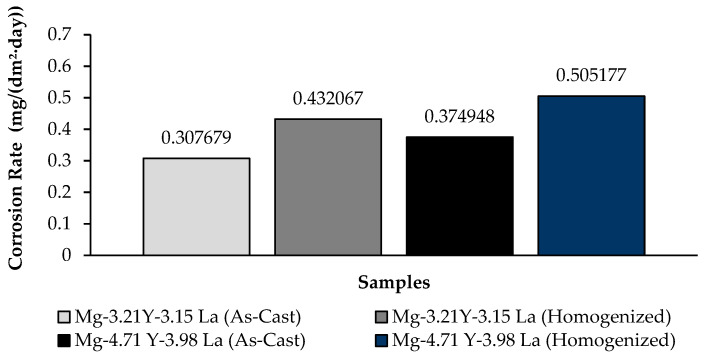
Results of immersion corrosion rates of Mg-3.21Y-3.15 La and Mg-4.71 Y-3.98 La alloys after casting and homogenization.

**Figure 9 materials-16-05141-f009:**
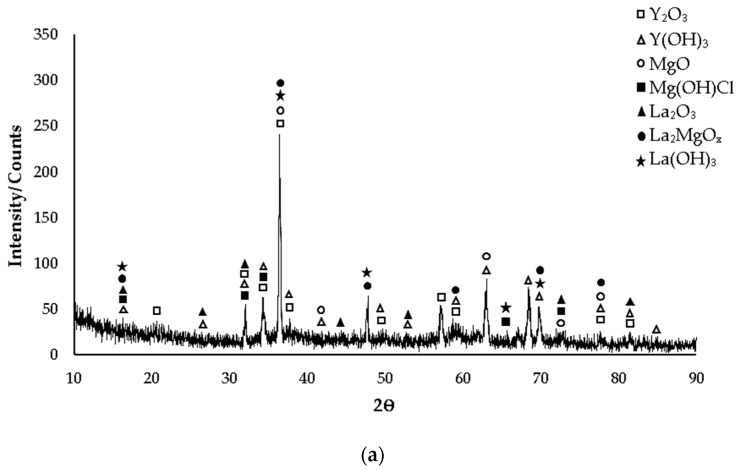
XRD models of after casting corroded alloys: (**a**) Mg-3.21Y-3.15 La and (**b**) Mg-4.71 Y-3.98 La.

**Figure 10 materials-16-05141-f010:**
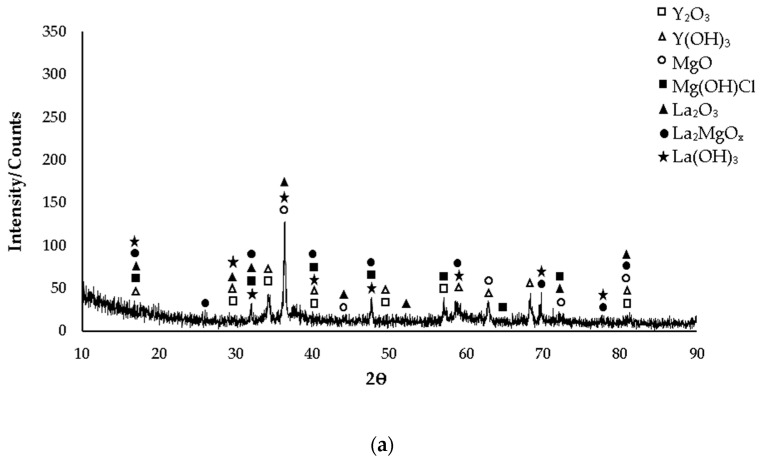
XRD models of alloys corroded after homogenization: (**a**) Mg-3.21Y-3.15 La and (**b**) Mg-4.71 Y-3.98 La.

**Figure 11 materials-16-05141-f011:**
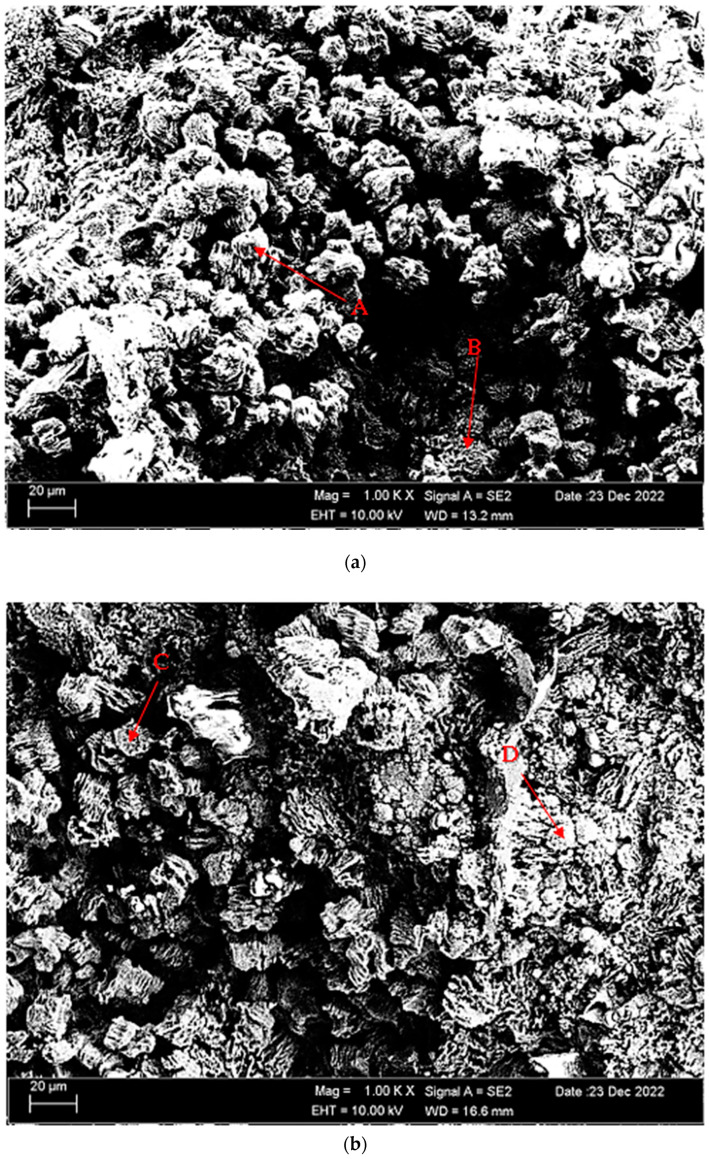
SEM micrographs of after casting corroded alloys (immersion corrosion): (**a**) Mg-3.21Y-3.15 La and (**b**) Mg-4.71 Y-3.98 La.

**Figure 12 materials-16-05141-f012:**
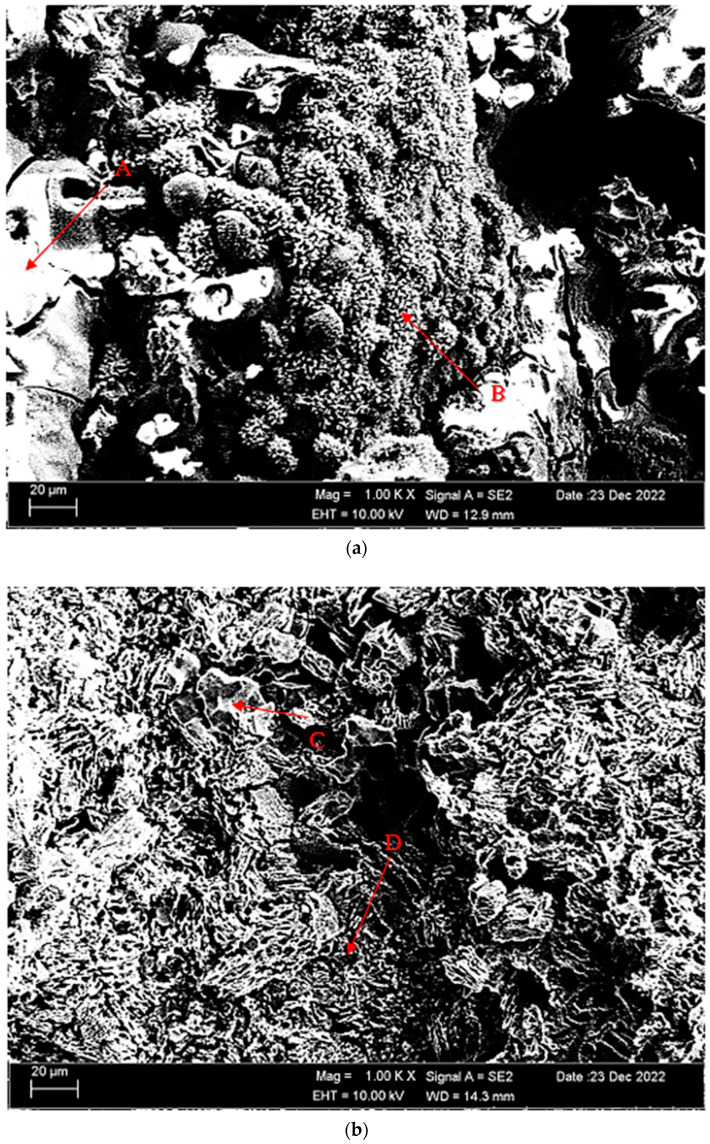
SEM micrographs of alloys corroded after homogenization (immersion corrosion): (**a**) Mg-3.21Y-3.15 La and (**b**) Mg-4.71 Y-3.98 La.

**Figure 13 materials-16-05141-f013:**
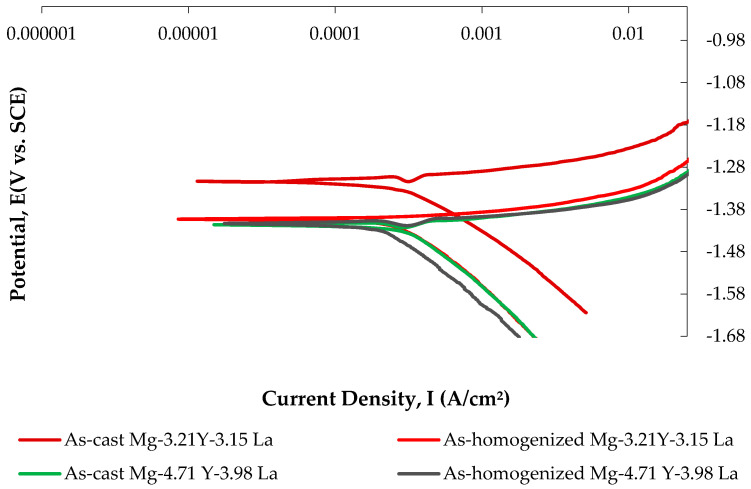
Potentiodynamic current–potential curves of Mg-3.21Y-3.15 La and Mg-4.71 Y-3.98 La alloys after casting and homogenization.

**Figure 14 materials-16-05141-f014:**
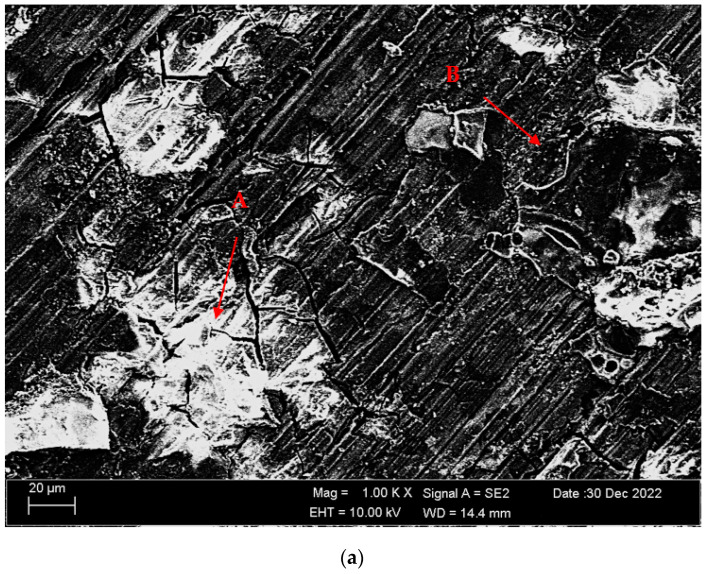
SEM micrographs of after casting corroded alloys (potentiodynamic polarization corrosion): (**a**) Mg-3.21Y-3.15 La and (**b**) Mg-4.71 Y-3.98 La.

**Figure 15 materials-16-05141-f015:**
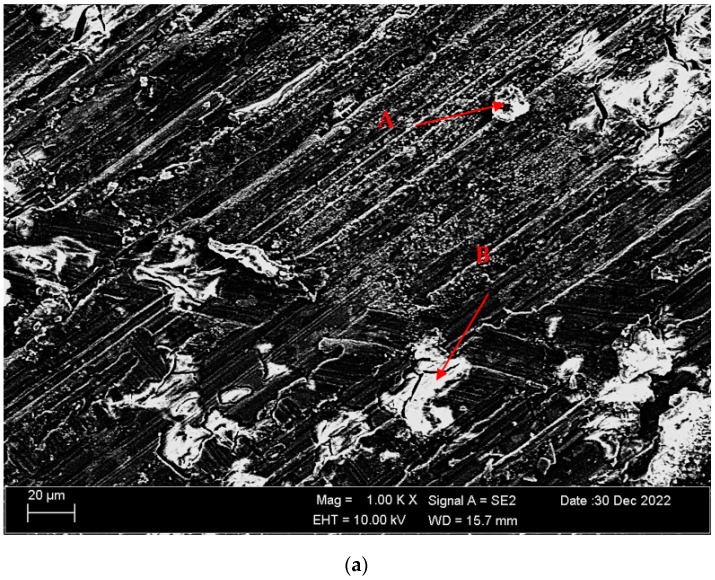
SEM micrographs of alloys corroded after homogenization (potentiodynamic polarization corrosion): (**a**) Mg-3.21Y-3.15 La and (**b**) Mg-4.71 Y-3.98 La.

**Table 1 materials-16-05141-t001:** Chemical compositions of the produced alloys (wt%).

	Chemical Composition (wt.%)			
Alloy	Y	La	Zr	Mg
Mg-3.21 Y-3.15 La	3.216	3.151	0.626	Bal.
Mg-4.71 Y-3.98 La	4.715	3.984	1.163	Bal.

**Table 2 materials-16-05141-t002:** EDS findings of Figure 4a,b phases (wt.%).

Points	Mg	Y	La	Zr
A	93.80	4.41	1.71	0.09
B	71.53	20.19	8.04	0.24
C	80.33	8.39	11.09	0.20
D	80.84	4.71	13.98	0.48
E	75.99	17.55	6.21	0.26
F	95.66	2.08	1.80	0.46

**Table 3 materials-16-05141-t003:** EDS findings of Figure 5a,b (wt.%).

Points	Mg	Y	La	Zr
A	79.55	12.43	7.42	0.59
B	80.93	7.10	11.62	0.34
C	88.01	6.39	5.36	0.25
D	83.42	8.15	7.91	0.53
E	85.10	4.94	9.63	0.33
F	81.63	7.10	7.58	3.69

**Table 4 materials-16-05141-t004:** EDS findings of Figure 11a,b (wt.%).

Points	Mg	Y	La	Zr	H	O	Cl	Na
A	76.80	2.24	0.70	0.31	0.77	18.62	0.45	0.13
B	66.62	1.35	1.24	0.51	0.30	29.13	0.75	0.11
C	75.72	5.52	7.20	0.37	2.22	8.49	0.12	0.36
D	64.32	3.65	5.90	-	0.28	25.45	-	0.39

**Table 5 materials-16-05141-t005:** EDS findings of Figure 12a,b (wt.%).

Points	Mg	Y	La	Zr	H	O	Cl	Na
A	39.13	3.09	5.07	0.08	0.63	51.14	0.39	0.47
B	27.35	1.36	4.48	0.11	0.32	65.35	0.33	0.74
C	69.58	2.38	6.09	0.19	-	21.04	0.10	0.63
D	78.82	2.28	4.59	0.45	2.03	11.31	0.02	0.50

**Table 6 materials-16-05141-t006:** Electrochemical corrosion data results of indicated in Figure 13.

	As-Cast		As-Homogenized	
Alloys	E_corr_ (V)	I_corr_ (μA/cm^2^)	E_corr_ (V)	I_corr_ (μA/cm^2^)
Mg-3.21Y-3.15 La	−1.17 ± 0.04	220 ± 1.5	−1.27 ± 0.04	294 ± 1.3
Mg-4.71 Y-3.98 La	−1.29 ± 0.01	357 ± 2.0	−1.30 ± 0.02	371 ± 1.2

**Table 7 materials-16-05141-t007:** EDS findings of Figure 14a,b (wt.%).

Points	Mg	Y	La	Zr	H	O	Cl	Na
A	55.29	4.10	4.05	0.20	1.22	34.28	0.79	0.08
B	50.94	2.16	2.58	0.10	1.80	41.79	0.49	0.14
C	53.04	1.99	1.71	0.87	-	42.26	0.03	0.11
D	60.27	2.00	2.11	-	1.27	33.45	0.08	0.82

**Table 8 materials-16-05141-t008:** EDS findings of Figure 15a,b (wt.%).

Points	Mg	Y	La	Zr	H	O	Cl	Na
A	39.25	3.59	4.04	-	3.54	47.99	0.74	0.83
B	50.51	3.37	2.04	0.18	1.60	41.71	0.03	0.30
C	46.77	0.19	1.50	0.44	0.29	50.65	0.10	0.06
D	34.48	0.49	1.30	-	0.11	63.39	-	0.22

## Data Availability

Not applicable.

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
