# Peer review of "Effect of Rare Earth Elements (Y, La) on Microstructural Characterization and Corrosion Behavior of Ternary Mg-Y-La Alloys"

_materials, 2023, doi:10.3390/ma16145141_

Round 1

Reviewer 1 Report

Notes on the article of Mohamed Ali Ibrahim Alwakwak, Ismail Esen, Hayrettin Ahlatcı and Esma Keskin «Effect of RE (Y, La) on Microstructural Characterization and Corrosion Behavior of Ternary Mg-Y-La Alloys»

The paper reports about the microstructure and corrosion behavior of magnesium alloys with minor addition of RE (Y, La). The authors showed that the eutectic compounds (α-Mg+Mg) RE (La/Y)) were formed in the alloys structure. The formation of crater structure was occurred on the surface of alloys samples after corrosion tests. The homogenization led to decrease of the corrosion resistance of both Mg-3.21Y-3.15 La and Mg-4.71 Y-3.98 La alloys. The article does not have a big theoretical importance. This is a typical work with easily predictable results. But it is quite interesting report using practical importance, which could be published after revisions that are listed below:

1. P.1, P 11-12. “In this study, the microstructural properties and corrosion behavior of RE (Y, La) added to magnesium in varying minor proportions after casting and homogenization heat treatment”. Was a verb missing here?

2. You wrote: “The phase structure was investigated with an optical microscope”. But you also give SEM micrographs. Please, give more details.

3. “Each sample's surface area was calculated using a perfect balance”. What does it mean?

4. How were corrosion rate (mg/dm2*day) and weight loss (mg/dm2) determined? Give the details of experiment (including of procedure of cleaning to remove the corrosion products and type of used electronic scale with its accuracy).

5. Give the details of determination of open circuit potential (OCP).

6. How many repetitions were done to determine the corrosion characteristics?

7. “Potentiodynamic polarization corrosion tests 3/4 cylindrical specimens wrapped in copper wire and cold mounted”. Could this way of preparing the samples affect the final results?

8. “Figure 2. After-cast alloy optical microscope…”, “Figure 4. As-cast alloy SEM micrographs…”. Use the same terminology.

9. What is the reason for using both optical and SEM microscopy if they provide the same data concerning microstructure? It is logically to use only SEM micrographs.

10. Does the fraction of phases decrease after homogenization?

11. “When the hardness results are examined in general, an increase in hardness is observed after the homogenization heat treatment”. According to Figure 6, this statement is false.

12. What is the reason to use mg/dm2 and mg/(dm2*day) instead of mg/сm2 and mg/(сm2*day)?

13. The paragraph describing the results of the immersion tests should be rewritten to take into account the experimental error. The authors draw conclusions about the best/worst corrosion resistance by comparing values that are the same with an error.

14. Where do the MgCO3 and the carbon contamination come from?

15. Table 6. Provide the experimental error.

16. Conclusions 2, 3 and 5 should be rewritten taking into account the values of the experimental error

17. Typos should be corrected:

- P. 2, line 43. It should be written “1.74 g/cm3” instead of “1.74 g/cm3”.

- P. 3, line 111. It should be written “CO2 + 0.8 SF6” instead of “CO2+ 0,8 SF6”.

- P. 3, line 127. It should be written “SiO2” instead of “SiO2”.

- P. 4, line 139. It should be written “1 mVs-1” instead of “1 mVs-1”.

- P. 11, lines 244 and 245. It should be written “mg/dm2” instead of “mg/dm2”.

- P. 14, line 289. It should be written “MgCO3” instead of “MgCO3”.

- P. 15, line 304. It should be written “La2O3” instead of “La2O3”.

- P. 15, line 305. It should be written “La2MgOx” instead of “La2MgOx”.

- P. 15, line 305. It should be written “La(OH)3” instead of “La(OH)3”.

- P. 20, line 407. It should be written “La2O3, and Y2O3” instead of “La2O3, and Y2O3”.

- P. 23, line 437. It should be written “Y2O3” instead of “Y2O3”.

Authors should carefully double-check the text of the article.

Reviewer 2 Report

1) Please write the full form of RE in the paper title

2) Please mention the novelty of the present work and elaborate how this is different from works of similar kind.

3)  mention the intended use and applications of this particular alloy

4) What is the size of RE particles used for alloying

5) Immersion test is supposed to be minimum of 1 week. In the current study only 72 hour immersion test done. Any reason for this

The language is good to go

Round 2

Reviewer 1 Report

The authors carefully revised the text of the article and answered all comments. The article should be accepted after minor revisions:

- Table 6. Error values should be written using a dot instead of a comma.

- Conclusion #5. See comment above.
